# A* Sampling

**Chris J. Maddison**
Dept. of Computer Science
University of Toronto
cmaddis@cs.toronto.edu

**Daniel Tarlow, Tom Minka**
Microsoft Research
{dtarlow,minka}@microsoft.com

## Abstract

The problem of drawing samples from a discrete distribution can be converted into a discrete optimization problem [1, 2, 3, 4]. In this work, we show how sampling from a continuous distribution can be converted into an optimization problem over continuous space. Central to the method is a stochastic process recently described in mathematical statistics that we call the *Gumbel process*. We present a new construction of the Gumbel process and A* *Sampling*, a practical generic sampling algorithm that searches for the maximum of a Gumbel process using A* search. We analyze the correctness and convergence time of A* Sampling and demonstrate empirically that it makes more efficient use of bound and likelihood evaluations than the most closely related adaptive rejection sampling-based algorithms.

## 1   Introduction

Drawing samples from arbitrary probability distributions is a core problem in statistics and machine learning. Sampling methods are used widely when training, evaluating, and predicting with probabilistic models. In this work, we introduce a generic sampling algorithm that returns exact independent samples from a distribution of interest. This line of work is important as we seek to include probabilistic models as subcomponents in larger systems, and as we seek to build probabilistic modelling tools that are usable by non-experts; in these cases, guaranteeing the quality of inference is highly desirable. There are a range of existing approaches for exact sampling. Some are specialized to specific distributions [5], but exact generic methods are based either on (adaptive) rejection sampling [6, 7, 8] or Markov Chain Monte Carlo (MCMC) methods where convergence to the stationary distribution can be guaranteed [9, 10, 11].

This work approaches the problem from a different perspective. Specifically, it is inspired by an algorithm for sampling from a discrete distribution that is known as the Gumbel-Max trick. The algorithm works by adding independent Gumbel perturbations to each configuration of a discrete negative energy function and returning the argmax configuration of the perturbed negative energy function. The result is an exact sample from the corresponding Gibbs distribution. Previous work [1, 3] has used this property to motivate samplers based on optimizing random energy functions but has been forced to resort to approximate sampling due to the fact that in structured output spaces, exact sampling appears to require instantiating exponentially many Gumbel perturbations.

Our first key observation is that we can apply the Gumbel-Max trick without instantiating all of the (possibly exponentially many) Gumbel perturbations. The same basic idea then allows us to extend the Gumbel-Max trick to continuous spaces where there will be infinitely many independent perturbations. Intuitively, for any given random energy function, there are many perturbation values that are irrelevant to determining the argmax so long as we have an upper bound on their values. We will show how to instantiate the relevant ones and bound the irrelevant ones, allowing us to find the argmax — and thus an exact sample.

There are a number of challenges that must be overcome along the way, which are addressed in this work. First, what does it mean to independently perturb space in a way analogous to perturbations in the Gumbel-Max trick? We introduce the Gumbel process, a special case of a stochastic process recently defined in mathematical statistics [12], which generalizes the notion of perturbation

over space. Second, we need a method for working with a Gumbel process that does not require instantiating infinitely many random variables. This leads to our novel construction of the Gumbel process, which draws perturbations according to a top-down ordering of their values. Just as the stick breaking construction of the Dirichlet process gives insight into algorithms for the Dirichlet process, our construction gives insight into algorithms for the Gumbel process. We demonstrate this by developing A* sampling, which leverages the construction to draw samples from arbitrary continuous distributions. We study the relationship between A* sampling and adaptive rejection sampling-based methods and identify a key difference that leads to more efficient use of bound and likelihood computations. We investigate the behaviour of A* sampling on a variety of illustrative and challenging problems.

## 2  The Gumbel Process

The Gumbel-Max trick is an algorithm for sampling from a categorical distribution over classes $i \in \{1, \ldots, n\}$ with probability proportional to $\exp(\phi(i))$. The algorithm proceeds by adding independent Gumbel-distributed noise to the log-unnormalized mass $\phi(i)$ and returns the optimal class of the perturbed distribution. In more detail, $G \sim \text{Gumbel}(m)$ is a Gumbel with location $m$ if $\text{P}(G \leq g) = \exp(-\exp(-g + m))$. The Gumbel-Max trick follows from the structure of Gumbel distributions and basic properties of order statistics; if $G(i)$ are i.i.d. $\text{Gumbel}(0)$, then $\text{argmax}_i \{G(i) + \phi(i)\} \sim \exp(\phi(i)) / \sum_i \exp(\phi(i))$. Further, for any $B \subseteq \{1, \ldots, n\}$

$$\max_{i \in B} \{G(i) + \phi(i)\} \sim \text{Gumbel}\left(\log \sum_{i \in B} \exp(\phi(i))\right) \tag{1}$$

$$\underset{i \in B}{\text{argmax}} \{G(i) + \phi(i)\} \sim \frac{\exp(\phi(i))}{\sum_{i \in B} \exp(\phi(i))} \tag{2}$$

Eq. 1 is known as *max-stability*—the highest order statistic of a sample of independent Gumbels also has a Gumbel distribution with a location that is the log partition function [13]. Eq. 2 is a consequence of the fact that Gumbels satisfy Luce's choice axiom [14]. Moreover, the max and argmax are independent random variables, see Appendix for proofs.

We would like to generalize the interpretation to continuous distributions as maximizing over the perturbation of a density $p(x) \propto \exp(\phi(x))$ on $\mathbb{R}^d$. The perturbed density should have properties analogous to the discrete case, namely that the max in $B \subseteq \mathbb{R}^d$ should be distributed as $\text{Gumbel}(\log \int_{x \in B} \exp(\phi(x)))$ and the distribution of the argmax in $B$ should be distributed $\propto \mathbf{1}(x \in B) \exp(\phi(x))$. The Gumbel process is a generalization satisfying these properties.

**Definition 1.** *Adapted from [12]. Let $\mu(B)$ be a sigma-finite measure on sample space $\Omega$, $B \subseteq \Omega$ measurable, and $G_\mu(B)$ a random variable. $\mathcal{G}_\mu = \{G_\mu(B) \mid B \subseteq \Omega\}$ is a Gumbel process, if*

1. *(marginal distributions) $G_\mu(B) \sim \text{Gumbel}(\log \mu(B))$.*

2. *(independence of disjoint sets) $G_\mu(B) \perp G_\mu(B^c)$.*

3. *(consistency constraints) for measurable $A, B \subseteq \Omega$, then*
$$G_\mu(A \cup B) = \max(G_\mu(A), G_\mu(B)).$$

The marginal distributions condition ensures that the Gumbel process satisfies the requirement on the max. The consistency requirement ensures that a realization of a Gumbel process is consistent across space. Together with the independence these ensure the argmax requirement. In particular, if $G_\mu(B)$ is the optimal value of some perturbed density restricted to $B$, then the event that the optima over $\Omega$ is contained in $B$ is equivalent to the event that $G_\mu(B) \geq G_\mu(B^c)$. The conditions ensure that $\text{P}(G_\mu(B) \geq G_\mu(B^c))$ is a probability measure proportional to $\mu(B)$ [12]. Thus, we can use the Gumbel process for a continuous measure $\mu(B) = \int_{x \in B} \exp(\phi(x))$ on $\mathbb{R}^d$ to model a perturbed density function where the optimum is distributed $\propto \exp(\phi(x))$. Notice that this definition is a generalization of the finite case; if $\Omega$ is finite, then the collection $\mathcal{G}_\mu$ corresponds exactly to maxes over subsets of independent Gumbels.

## 3  Top-Down Construction for the Gumbel Process

While [12] defines and constructs a general class of stochastic processes that include the Gumbel process, the construction that proves their existence gives little insight into how to execute a con-

tinuous version of the Gumbel-Max trick. Here we give an alternative algorithmic construction that will form the foundation of our practical sampling algorithm. In this section we assume $\log \mu(\Omega)$ can be computed tractably; this assumption will be lifted in Section 4. To explain the construction, we consider the discrete case as an introductory example.

Suppose $G_\mu(i) \sim \mathrm{Gumbel}(\phi(i))$ is a set of independent Gumbel random variables for $i \in \{1, \ldots, n\}$. It would be straightforward to sample the variables then build a heap of the $G_\mu(i)$ values and also have heap nodes store the index $i$ associated with their value. Let $B_i$ be the set of indices that appear in the subtree rooted at the node with index $i$. A property of the heap is that the root $(G_\mu(i), i)$ pair is the max and argmax of the set of Gumbels with index in $B_i$. The key idea of our construction is to sample the independent set of random variables by instantiating this heap from root to leaves. That is, we will first sample the root node, which is the global max and argmax, then we will recurse, sampling the root's two children conditional upon the root. At the end, we

---

**Algorithm 1** Top-Down Construction

**input** sample space $\Omega$, measure $\mu(B) = \int_B \exp(\phi) dm$
  $(B_1, Q) \leftarrow (\Omega, \mathrm{Queue})$
  $G_1 \sim \mathrm{Gumbel}(\log \mu(\Omega))$
  $X_1 \sim \exp(\phi(x))/\mu(\Omega)$
  $Q.push(1)$
  $k \leftarrow 1$
  **while** $!Q.empty()$ **do**
    $p \leftarrow Q.pop()$
    $L, R \leftarrow partition(B_p - \{X_p\})$
    **for** $C \in \{L, R\}$ **do**
      **if** $C \neq \emptyset$ **then**
        $k \leftarrow k + 1$
        $B_k \leftarrow C$
        $G_k \sim \mathrm{TruncGumbel}(\log \mu(B_k), G_p)$
        $X_k \sim \mathbf{1}(x \in B_k) \exp(\phi(x))/\mu(B_k)$
        $Q.push(k)$
        **yield** $(G_k, X_k)$

---

will have sampled a heap full of values and indices; reading off the value associated with each index will yield a draw of independent Gumbels from the target distribution.

We sketch an inductive argument. For the base case, sample the max and its index $i^*$ using their distributions that we know from Eq. 1 and Eq. 2. Note the max and argmax are independent. Also let $B_{i^*} = \{0, \ldots, n-1\}$ be the set of all indices. Now, inductively, suppose have sampled a partial heap and would like to recurse downward starting at $(G_\mu(p), p)$. Partition the remaining indices to be sampled $B_p - \{p\}$ into two subsets $L$ and $R$ and let $l \in L$ be the left argmax and $r \in R$ be the right argmax. Let $[\geq p]$ be the indices that have been sampled already. Then

$$p\left(G_\mu(l) = g_l, G_\mu(r) = g_r, \{G_\mu(k) = g_k\}_{k \in [\geq p]} \,|\, [\geq p]\right) \tag{3}$$

$$\propto p\left(\max_{i \in L} G_\mu(i) = g_l\right) p\left(\max_{i \in R} G_\mu(i) = g_r\right) \prod_{k \in [\geq p]} p_k(G_\mu(k) = g_k) \mathbf{1}\left(g_k \geq g_{\mathcal{L}(k)} \wedge g_k \geq g_{\mathcal{R}(k)}\right)$$

where $\mathcal{L}(k)$ and $\mathcal{R}(k)$ denote the left and right children of $k$ and the constraints should only be applied amongst nodes $[\geq p] \cup \{l, r\}$. This implies

$$p\left(G_\mu(l) = g_l, G_\mu(r) = g_r \,|\, \{G_\mu(k) = g_k\}_{k \in [\geq p]}, [\geq p]\right)$$

$$\propto p\left(\max_{i \in L} G_\mu(i) = g_l\right) p\left(\max_{i \in R} G_\mu(i) = g_r\right) \mathbf{1}(g_p > g_l) \mathbf{1}(g_p > g_r). \tag{4}$$

Eq. 4 is the joint density of two independent Gumbels truncated at $G_\mu(p)$. We could sample the children maxes and argmaxes by sampling the independent Gumbels in $L$ and $R$ respectively and computing their maxes, rejecting those that exceed the known value of $G_\mu(p)$. Better, the truncated Gumbel distributions can be sampled efficiently via CDF inversion[1], and the independent argmaxes within $L$ and $R$ can be sampled using Eq. 2. Note that any choice of partitioning strategy for $L$ and $R$ leads to the same distribution over the set of Gumbel values.

The basic structure of this top-down sampling procedure allows us to deal with infinite spaces; we can still generate an infinite descending heap of Gumbels and locations as if we had made a heap from an infinite list. The algorithm (which appears as Algorithm 1) begins by sampling the optimal value $G_1 \sim \mathrm{Gumbel}(\log \mu(\Omega))$ over sample space $\Omega$ and its location $X_1 \sim \exp(\phi(x))/\mu(\Omega)$. $X_1$ is removed from the sample space and the remaining sample space is partitioned into $L$ and $R$. The optimal Gumbel values for $L$ and $R$ are sampled from a Gumbel with location log measure of their

respective sets, but truncated at $G_1$. The locations are sampled independently from their sets, and the procedure recurses. As in the discrete case, this yields a stream of $(G_k, X_k)$ pairs, which we can think of as being nodes in a heap of the $G_k$'s.

If $G_\mu(x)$ is the value of the perturbed negative energy at $x$, then Algorithm 1 instantiates this function at countably many points by setting $G_\mu(X_k) = G_k$. In the discrete case we eventually sample the complete perturbed density, but in the continuous case we simply generate an infinite stream of locations and values. The sense in which Algorithm 1 constructs a Gumbel process is that the collection $\{\max\{G_k \,|\, X_k \in B\} \,|\, B \subseteq \Omega\}$ satisfies Definition 1. The intuition should be provided by the introductory argument; a full proof appears in the Appendix. An important note is that because $G_k$'s are sampled in descending order along a path in the tree, when the first $X_k$ lands in set $B$, the value of $\max\{G_k \,|\, X_k \in B\}$ will not change as the algorithm continues.

## 4 A* Sampling

The Top-Down construction is not executable in general, because it assumes $\log \mu(\Omega)$ can be computed efficiently. A* sampling is an algorithm that executes the Gumbel-Max trick without this assumption by exploiting properties of the Gumbel process. Henceforth A* sampling refers exclusively to the continuous version.

A* sampling is possible because we can transform one Gumbel process into another by adding the difference in their log densities. Suppose we have two continuous measures $\mu(B) = \int_{x \in B} \exp(\phi(x))$ and $\nu(B) = \int_{x \in B} \exp(i(x))$. Let pairs $(G_k, X_k)$ be draws from the Top-Down construction for $\mathcal{G}_\nu$. If $o(x) = \phi(x) - i(x)$ is bounded, then we can recover $\mathcal{G}_\mu$ by adding the difference $o(X_k)$ to every $G_k$; i.e., $\{\max\{G_k + o(X_k) \,|\, X_k \in B\} \,|\, B \subseteq \mathbb{R}^d\}$ is a Gumbel process with measure $\mu$. As an example, if $\nu$ were a prior and $o(x)$ a bounded log-likelihood, then we could simulate the Gumbel process corresponding to the posterior by adding $o(X_k)$ to every $G_k$ from a run of the construction for $\nu$.

This "linearity" allows us to decompose a target log density function into a tractable $i(x)$ and boundable $o(x)$. The tractable component is analogous to the proposal distribution in a rejection sampler. A* sampling searches for $\arg\max\{G_k + o(X_k)\}$ within the heap of $(G_k, X_k)$ pairs from the Top-Down construction of $\mathcal{G}_\nu$. The search is an A* procedure: nodes in the search tree correspond to increasingly refined regions in space, and the search is guided by upper and lower bounds that are computed for each region. Lower bounds for

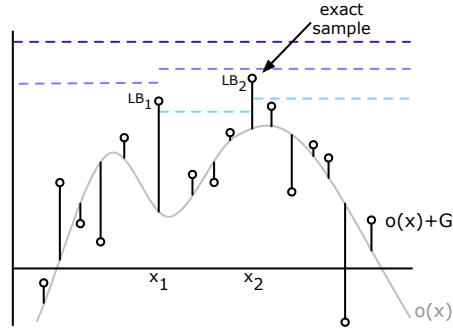

Figure 1: Illustration of A* sampling.

---

**Algorithm 2** A* Sampling

---

**input** log density $i(x)$, difference $o(x)$, bounding function $M(B)$, and $partition$
  $(LB, \, X^*, \, k) \leftarrow (-\infty, \, \text{null}, \, 1)$
  $Q \leftarrow \text{PriorityQueue}$
  $G_1 \sim \text{Gumbel}(\log \nu(\mathbb{R}^d))$
  $X_1 \sim \exp(i(x))/\nu(\mathbb{R}^d)$
  $M_1 \leftarrow M(\mathbb{R}^d)$
  $Q.pushWithPriority(1, G_1 + M_1)$
  **while** $!Q.empty()$ and $LB < Q.topPriority()$ **do**
    $p \leftarrow Q.popHighest()$
    $LB_p \leftarrow G_p + o(X_p)$
    **if** $LB < LB_p$ **then**
      $LB \leftarrow LB_p$
      $X^* \leftarrow X_p$
    $L, R \leftarrow partition(B_p, X_p)$
    **for** $C \in \{L, R\}$ **do**
      **if** $C \neq \emptyset$ **then**
        $k \leftarrow k + 1$
        $B_k \leftarrow C$
        $G_k \sim \text{TruncGumbel}(\log \nu(B_k), G_p)$
        $X_k \sim \mathbf{1}(x \in B_k)\exp(i(x))/\nu(B_k)$
        **if** $LB < G_k + M_p$ **then**
          $M_k \leftarrow M(B_k)$
          **if** $LB < G_k + M_k$ **then**
            $Q.pushWithPriority(k, G_k + M_k)$
**output** $(LB, X^*)$

---

region $B$ come from drawing the max $G_k$ and argmax $X_k$ of $\mathcal{G}_\nu$ within $B$ and evaluating $G_k + o(X_k)$. Upper bounds come from the fact that

$$\max\{G_k + o(X_k) \,|\, X_k \in B\} \leq \max\{G_k \,|\, X_k \in B\} + M(B),$$

where $M(B)$ is a bounding function for a region, $M(B) \geq o(x)$ for all $x \in B$. $M(B)$ is not random and can be implemented using methods from e.g., convex duality or interval analysis. The first term on the RHS is the $G_k$ value used in the lower bound.

The algorithm appears in Algorithm 2 and an execution is illustrated in Fig. 1. The algorithm begins with a global upper bound (dark blue dashed). $G_1$ and $X_1$ are sampled, and the first lower bound $LB_1 = G_1 + o(X_1)$ is computed. Space is split, upper bounds are computed for the new children regions (medium blue dashed), and the new nodes are put on the queue. The region with highest upper bound is chosen, the maximum Gumbel in the region, $(G_2, X_2)$, is sampled, and $LB_2$ is computed. The current region is split at $X_2$ (producing light blue dashed bounds), after which $LB_2$ is greater than the upper bound for any region on the queue, so $LB_2$ is guaranteed to be the max over the infinite tree of $G_k + o(X_k)$. Because $\max\{G_k + o(X_k) \mid X_k \in B\}$ is a Gumbel process with measure $\mu$, this means that $X_2$ is an exact sample from $p(x) \propto \exp(\phi(x))$ and $LB_2$ is an exact sample from $\mathrm{Gumbel}(\log \mu(\mathbb{R}^d))$. Proofs of termination and correctness are in the Appendix.

**A\* Sampling Variants.** There are several variants of A\* sampling. When more than one sample is desired, bound information can be reused across runs of the sampler. In particular, suppose we have a partition of $\mathbb{R}^d$ with bounds on $o(x)$ for each region. A\* sampling could use this by running a search independently for each region and returning the max Gumbel. The maximization can be done lazily by using A\* search, only expanding nodes in regions that are needed to determine the global maximum. The second variant trades bound computations for likelihood computations by drawing more than one sample from the auxiliary Gumbel process at each node in the search tree. In this way, more lower bounds are computed (costing more likelihood evaluations), but if this leads to better lower bounds, then more regions of space can be pruned, leading to fewer bound evaluations. Finally, an interesting special case of A\* sampling can be implemented when $o(x)$ is unimodal in 1D. In this case, at every split of a parent node, one child can immediately be pruned, so the "search" can be executed without a queue. It simply maintains the currently active node and drills down until it has provably found the optimum.

## 5    Comparison to Rejection Samplers

Our first result relating A\* sampling to rejection sampling is that if the same global bound $M = M(\mathbb{R}^d)$ is used at all nodes within A\* sampling, then the runtime of A\* sampling is equivalent to that of standard rejection sampling. That is, the distribution over the number of iterations is distributed as a Geometric distribution with rate parameter $\mu(\mathbb{R}^d)/(\exp(M)\nu(\mathbb{R}^d))$. A proof is in the Appendix as part of the proof of termination.

When bounds are refined, A\* sampling bears similarity to adaptive rejection sampling-based algorithms. In particular, while it appears only to have been applied in discrete domains, OS\* [7] is a general class of adaptive rejection sampling methods that maintain piecewise bounds on the target distribution. If piecewise constant bounds are used (henceforth we assume OS\* uses only constant bounds) the procedure can be described as follows: at each step, (1) a region $B$ with bound $U(B)$ is sampled with probability proportional to $\nu(B) \exp(M(B))$, (2) a point is drawn from the proposal distribution restricted to the chosen region; (3) standard accept/rejection computations are performed using the regional bound, and (4) if the point is rejected, a region is chosen to be split into two, and new bounds are computed for the two regions that were created by the split. This process repeats until a point is accepted.

Steps (2) and (4) are performed identically in A\* when sampling argmax Gumbel locations and when splitting a parent node. A key difference is how regions are chosen in step (1). In OS\*, a region is drawn according to volume of the region under the proposal. Note that piece selection could be implemented using the Gumbel-Max trick, in which case we would choose the piece with maximum $G_B + M(B)$ where $G_B \sim \mathrm{Gumbel}(\log \nu(B))$. In A\* sampling the region with highest upper bound is chosen, where the upper bound is $G_B + M(B)$. The difference is that $G_B$ values are reset after each rejection in OS\*, while they persist in A\* sampling until a sample is returned.

The effect of the difference is that A\* sampling more tightly couples together where the accepted sample will be and which regions are refined. Unlike OS\*, it can go so far as to prune a region from the search, meaning there is zero probability that the returned sample will be from that region, and that region will never be refined further. OS\*, on the other hand, is blind towards where the sample that will eventually be accepted comes from and will on average waste more computation refining regions that ultimately are not useful in drawing the sample. In experiments, we will see that A\* consistently dominates OS\*, refining the function less while also using fewer likelihood evaluations. This is possible because the persistence inside A\* sampling focuses the refinement on the regions that are important for accepting the current sample.

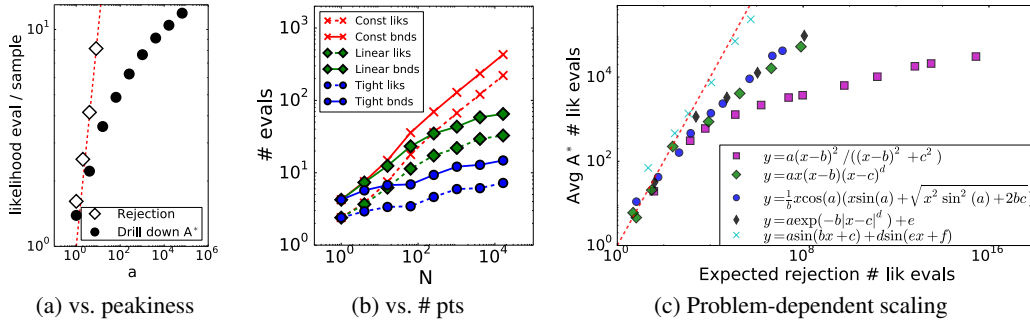

(a) vs. peakiness          (b) vs. # pts          (c) Problem-dependent scaling

Figure 2: (a) Drill down algorithm performance on $p(x) = \exp(-x)/(1+x)^a$ as function of $a$. (b) Effect of different bounding strategies as a function of number of data points; number of likelihood and bound evaluations are reported. (c) Results of varying observation noise in several nonlinear regression problems.

## 6  Experiments

There are three main aims in this section. First, understand the empirical behavior of A* sampling as parameters of the inference problem and $o(x)$ bounds vary. Second, demonstrate generality by showing that A* sampling algorithms can be instantiated in just a few lines of model-specific code by expressing $o(x)$ symbolically, and then using a branch and bound library to automatically compute bounds. Finally, compare to OS* and an MCMC method (slice sampling). In all experiments, regions in the search trees are hyper rectangles (possibly with infinite extent); to split a region $A$, choose the dimension with the largest side length and split the dimension at the sampled $X_k$ point.

### 6.1  Scaling versus Peakiness and Dimension

In the first experiment, we sample from $p(x) = \exp(-x)/(1+x)^a$ for $x > 0, a > 0$ using $\exp(-x)$ as the proposal distribution. In this case, $o(x) = -a \log(1+x)$ which is unimodal, so the drill down variant of A* sampling can be used. As $a$ grows, the function becomes peakier; while this presents significant difficulty for vanilla rejection sampling, the cost to A* is just the cost of locating the peak, which is essentially binary search. Results averaged over 1000 runs appear in Fig. 2 (a).

In the second experiment, we run A* sampling on the clutter problem [15], which estimates the mean of a fixed covariance isotropic Gaussian under the assumption that some points are outliers. We put a Gaussian prior on the inlier mean and set $i(x)$ to be equal to the prior, so $o(x)$ contains just the likelihood terms. To compute bounds on the total log likelihood, we compute upper bounds on the log likelihood of each point independently then sum up these bounds. We will refer to these as "constant" bounds. In $D$ dimensions, we generated 20 data points with half within $[-5, -3]^D$ and half within $[2, 4]^D$, which ensures that the posterior is sharply bimodal, making vanilla MCMC quickly inappropriate as $D$ grows. The cost of drawing an exact sample as a function of $D$ (averaged over 100 runs) grows exponentially in $D$, but the problem remains reasonably tractable as $D$ grows ($D = 3$ requires 900 likelihood evaluations, $D = 4$ requires 4000). The analogous OS* algorithm run on the same set of problems requires 16% to 40% more computation on average over the runs.

### 6.2  Bounding Strategies

Here we investigate alternative strategies for bounding $o(x)$ in the case where $o(x)$ is a sum of per-instance log likelihoods. To allow easy implementation of a variety of bounding strategies, we choose the simple problem of estimating the mean of a 1D Gaussian given $N$ observations. We use three types of bounds: constant bounds as in the clutter problem; linear bounds, where we compute linear upper bounds on each term of the sum, then sum the linear functions and take the max over the region; and quadratic bounds, which are the same as linear except quadratic bounds are computed on each term. In this problem, quadratic bounds are tight. We evaluate A* sampling using each of the bounding strategies, varying $N$. See Fig. 2 (b) for results.

For $N = 1$, all bound types are equivalent when each expands around the same point. For larger $N$, the looseness of each per-point bound becomes important. The figure shows that, for large $N$, using linear bounds multiplies the number of evaluations by 3, compared to tight bounds. Using constant bounds multiplies the number of evaluations by $O(\sqrt{N})$. The Appendix explains why this happens

and shows that this behavior is expected for any estimation problem where the width of the posterior shrinks with $N$.

## 6.3 Using Generic Interval Bounds

Here we study the use of bounds that are derived automatically by means of interval methods [16]. This suggests how A* sampling (or OS*) could be used within a more general purpose probabilistic programming setting. We chose a number of nonlinear regression models inspired by problems in physics, computational ecology, and biology. For each, we use FuncDesigner [17] to symbolically construct $o(x)$ and automatically compute the bounds needed by the samplers.

Several expressions for $y = f(x)$ appear in the legend of Fig. 2 (c), where letters $a$ through $f$ denote parameters that we wish to sample. The model in all cases is $y_n = f(x_n) + \epsilon_n$ where $n$ is the data point index and $\epsilon_n$ is Gaussian noise. We set uniform priors from a reasonable range for all parameters (see Appendix) and generated a small (N=3) set of training data from the model so that posteriors are multimodal. The peakiness of the posterior can be controlled by the magnitude of the observation noise; we varied this from large to small to produce problems over a range of difficulties. We use A* sampling to sample from the posterior five times for each model and noise setting and report the average number of likelihood evaluations needed in Fig. 2 (c) (y-axis). To establish the difficulty of the problems, we estimate the expected number of likelihood evaluations needed by a rejection sampler to accept a sample. The savings over rejection sampling is often exponentially large, but it varies per problem and is not necessarily tied to the dimension. In the example where savings are minimal, there are many symmetries in the model, which leads to uninformative bounds. We also compared to OS* on the same class of problems. Here we generated 20 random instances with a fixed intermediate observation noise value for each problem and drew 50 samples, resetting the bounds after each sample. The average cost (heuristically set to # likelihood evaluations plus 2 × # bound evaluations) of OS* for the five models in Fig. 2 (c) respectively was 21%, 30%, 11%, 21%, and 27% greater than for A*.

## 6.4 Robust Bayesian Regression

Here our aim is to do Bayesian inference in a robust linear regression model $y_n = \boldsymbol{w}^\mathsf{T}\boldsymbol{x}_n + \epsilon_n$ where noise $\epsilon_n$ is distributed as standard Cauchy and $\boldsymbol{w}$ has an isotropic Gaussian prior. Given a dataset $\mathcal{D} = \{\boldsymbol{x}_n, y_n\}_{n=1}^N$ our goal is to draw samples from the posterior $\mathrm{P}(\boldsymbol{w}\,|\,\mathcal{D})$. This is a challenging problem because the heavy-tailed noise model can lead to multimodality in the posterior over $\boldsymbol{w}$. The log likelihood is $\mathcal{L}(\boldsymbol{w}) = \sum_n \log(1 + (\boldsymbol{w}^\mathsf{T}\boldsymbol{x}_n - y_n)^2)$. We generated $N$ data points with input dimension $D$ in such a way that the posterior is bimodal and symmetric by setting $\boldsymbol{w}^* = [2, ..., 2]^\mathsf{T}$, generating $X' \sim \mathrm{randn}(N/2, D)$ and $y' \sim X'\boldsymbol{w}^* + .1 \times \mathrm{randn}(N/2)$, then setting $X = [X'; X']$ and $y = [y'; -y']$. There are then equally-sized modes near $\boldsymbol{w}^*$ and $-\boldsymbol{w}^*$. We decompose the posterior into a uniform $i(\cdot)$ within the interval $[-10, 10]^D$ and put all of the prior and likelihood terms into $o(\cdot)$. Bounds are computed per point; in some regions the per point bounds are linear, and in others they are quadratic. Details appear in the Appendix.

We compare to OS*, using two refinement strategies that are discussed in [7]. The first is directly analogous to A* sampling and is the method we have used in the earlier OS* comparisons. When a point is rejected, refine the piece that was proposed from at the sampled point, and split the dimension with largest side length. The second method splits the region with largest probability under the proposal. We ran experiments on several random draws of the data and report performance along the two axes that are the dominant costs: how many bound computations were used, and how many likelihood evaluations were used. To weigh the tradeoff between the two, we did a rough asymptotic calculation of the costs of bounds versus likelihood computations and set the cost of a bound computation to be $D + 1$ times the cost of a likelihood computation.

In the first experiment, we ask each algorithm to draw a single exact sample from the posterior. Here, we also report results for the variants of A* sampling and OS* that trade off likelihood computations for bound computations as discussed in Section 4. A representative result appears in Fig. 3 (left). Across operating points, A* consistently uses fewer bound evaluations and fewer likelihood evaluations than both OS* refinement strategies.

In the second experiment, we ask each algorithm to draw 200 samples from the posterior and experiment with the variants that reuse bound information across samples. A representative result appears in Fig. 3 (right). Here we see that the extra refinement done by OS* early on allows it to use fewer likelihood evaluations at the expense of more bound computations, but A* sampling operates at a

point that is not achievable by OS*. For all of these problems, we ran a random direction slice sampler [18] that was given 10 times the computational budget that A* sampling used to draw 200 samples. The slice sampler had trouble mixing when $D > 1$. Across the five runs for $D = 2$, the sampler switched modes once, and it did not ever switch modes when $D > 2$.

## 7    Discussion

This work answers a natural question: is there a Gumbel-Max trick for continuous spaces, and can it be leveraged to develop tractable algorithms for sampling from continuous distributions?

In the discrete case, recent work on "Perturb and MAP" (P&M) methods [1, 19, 2] that draw samples as the argmaxes of random energy functions has shown value in developing approximate, correlated perturbations. It is natural to think about continuous analogs in which exactness is abandoned in favor of more efficient computation. A question is if the approximations can be developed in a principled way, like how [3] showed a particular form of correlated discrete perturbation gives rise to bounds on the log partition function. Can analogous rigorous approximations be established in the continuous case? We hope this work is a starting point for exploring that question.

We do not solve the problem of high dimensions. There are simple examples where bounds become uninformative in high dimensions, such as when sampling a density that is

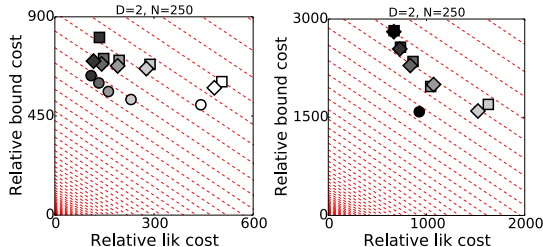

Figure 3: A* (circles) versus OS* (squares and diamonds) computational costs on Cauchy regression experiments of varying dimension. Square is refinement strategy that splits node where rejected point was sampled; Diamond refines region with largest mass under the proposal distribution. Red lines denote lines of equi-total computational cost and are spaced on a log scale by 10% increase increments. Color of markers denotes the rate of refinement, ranging from (darkest) refining for every rejection (for OS*) or one lower bound evaluation per node expansion (for A*) to (lightest) refining on 10% of rejections (for OS*) or performing $\text{Poisson}(\frac{1}{.1} - 1) + 1$ lower bound evaluations per node expansion (for A*). (left) Cost of drawing a single sample, averaged over 20 random data sets. (right) Drawing 200 samples averaged over 5 random data sets. Results are similar over a range of $N$'s and $D = 1, \ldots, 4$.

uniform over a hypersphere when using hyperrectangular search regions. In this case, little is gained over vanilla rejection sampling. An open question is if the split between $i(\cdot)$ and $o(\cdot)$ can be adapted to be node-specific during the search. An adaptive rejection sampler would be able to do this, which would allow leveraging parameter-varying bounds in the proposal distributions. This might be an important degree of freedom to exercise, particularly when scaling up to higher dimensions.

There are several possible follow-ons including the discrete version of A* sampling and evaluating A* sampling as an estimator of the log partition function. In future work, we would like to explore taking advantage of conditional independence structure to perform more intelligent search, hopefully helping the method scale to larger dimensions. Example starting points might be ideas from AND/OR search [20] or branch and bound algorithms that only branch on a subset of dimensions [21].

## Acknowledgments

This research was supported by NSERC. We thank James Martens and Radford Neal for helpful discussions, Elad Mezuman for help developing early ideas related to this work, and Roger Grosse for suggestions that greatly improved this work.

## Footnotes

[1]$G \sim \mathrm{TruncGumbel}(\phi, b)$ if $G$ has CDF $\exp(-\exp(-\min(g, b) + \phi))/\exp(-\exp(-b+\phi))$. To sample efficiently, return $G = -\log(\exp(-b - \gamma + \phi) - \log(U)) - \gamma + \phi$ where $U \sim \mathrm{uniform}[0, 1]$.

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
