[Supplementary Material · cameraready_appendix.pdf]

## Appendix for "A* Sampling"

In this appendix we prove the main theoretical results of the paper and provide additional experimental details. First, define the following shorthand

$$e_\phi(g) = \exp(-g + \phi)$$
$$F_\phi(g) = \exp(-\exp(-g + \phi))$$
$$f_\phi(g) = e_\phi(g)F_\phi(g)$$

Thus $F_\phi(g)$ is the CDF and $f_\phi(g)$ the PDF of a $\mathrm{Gumbel}(\phi)$. The following identities are easy to verify and will be reused throughout the appendix.

$$F_\phi(g)F_\gamma(g) = F_{\log(\exp(\phi)+\exp(\gamma))}(g) \tag{5}$$

$$\int_{x=a}^{b} e_\gamma(g)F_\phi(g) = (F_\phi(b) - F_\phi(a))\frac{\exp(\gamma)}{\exp(\phi)} \tag{6}$$

## Joint Distribution of Gumbel Max and Argmax

Suppose $G(i) \sim \mathrm{TruncGumbel}(\phi(i), b)$ are $n$ independent truncated Gumbels and $Z = \sum_{i=1}^{n} \exp(\phi(i))$, then we are interested in deriving the joint distribution of $i^* = \mathrm{argmax}_{i=1}^{n} G(i)$ and $G(i^*) = \max_{i=1}^{n} G(i)$.

$$
\begin{aligned}
p(k, g) &= p(i^* = k, \ G(i^*) = g) \\
&= p(G(k) = g, \ G(k) \geq \max_{i \neq k} G(i)) \\
&= \frac{f_{\phi(k)}(g)\mathbf{1}(g \leq b)}{F_{\phi(k)}(b)} \prod_{i \neq k} \frac{F_{\phi(i)}(g)}{F_{\phi(i)}(b)} \\
&= \exp(-g + \phi(k))\mathbf{1}(g \leq b) \prod_{i=1}^{n} \frac{F_{\phi(i)}(g)}{F_{\phi(i)}(b)} \\
&= \frac{\exp(\phi(k))}{Z} \exp(-g + \log Z)\mathbf{1}(g \leq b) \prod_{i=1}^{n} \frac{F_{\phi(i)}(g)}{F_{\phi(i)}(b)} \\
&= \frac{\exp(\phi(k))}{Z} \frac{f_{\log Z}(g)\mathbf{1}(g \leq b)}{F_{\log Z}(b)}
\end{aligned}
$$

This is the Gibbs distribution and the density of a $\mathrm{TruncGumbel}(\log Z, b)$. Thus, for any $B \subseteq \{1, \ldots, n\}$

$$\max_{i \in B} G(i) \sim \mathrm{TruncGumbel}\left(\log \sum_{i \in B} \exp(\phi(i)), \ b\right)$$

$$\mathrm{argmax}_{i \in B} G(i) \sim \frac{\mathbf{1}(i \in B)\exp(\phi(i))}{\sum_{i \in B} \exp(\phi(i))}$$

$$\max_{i \in B} G(i) \ \perp \ \mathrm{argmax}_{i \in B} G(i)$$

These results are well-known. The fact that max Gumbel value has a location that is the log partition function means we can use samples of it as an estimator of log partition functions with known variance $\pi^2/6N$ for $N$ samples [13]. Eq. 2 shows that Gumbels satisfy Luce's choice axiom [14]. In fact, it is a well-known result in random choice theory that the only distribution satisfying Eq. 2 is Gumbel. Notice that the argmax is also independent of the bound, and $b = \infty$ is a valid choice.

---

**Algorithm 3** In-Order Construction

---

**input** sample space $\Omega$, sigma-finite measure $\mu(B)$
   $G_1 \sim \text{Gumbel}(\log \mu(\Omega))$
   $X_1 \sim \exp(\phi(x))/\mu(\Omega)$
   $k \leftarrow 1$
   **while** $\Omega \neq \emptyset$ **do**
      $k \leftarrow k + 1$
      $\Omega \leftarrow \Omega - \{X_{k-1}\}$
      $G_k \sim \text{TruncGumbel}(\log \mu(\Omega), G_{k-1})$
      $X_k \sim \mathbf{1}(x \in \Omega)\exp(\phi(x))/\mu(\Omega)$
      **yeild** $(G_k, X_k)$

---

Figure 4: Visualization of a realization of a Gumbel process as produced by the Top-Down (Alg. 1) and In-Order (Alg. 3) constructions for the first few steps. Blue arrows indicate truncation. Black lines in $\Omega$ indicate partitioning for Algorithm 1. In particular $B_2 = B_4 \cup B_5$. Note the sense in which they are simply re-orderings of each other.

## Analysis of Top-Down Construction

### Correctness of the Top-Down Construction of the Gumbel Process

The goal of this section is to prove that the Top-Down Construction constructs the Gumbel process. In particular, we will argue if we run Algorithm 1 with $\mu$ on $\Omega$, then the collection

$$\mathcal{G}'_\mu = \{\max\{G_k \,|\, X_k \in B\} \,|\, B \subseteq \Omega\}$$

is a Gumbel process $\mathcal{G}'_\mu \stackrel{d}{=} \mathcal{G}_\mu$. In order to do this we consider a special case of Algorithm 1 in which space is not subdivided, $partition(B) = (B, \emptyset)$. In this case the construction takes on a particular simple form, since no queue is needed, see Algorithm 3. We call this special case the In-Order Construction, because is produces the Gumbel values in non-increasing order.

We proceed by arguing that subdividing space has no effect on the distribution of the top $n$ Gumbels. This means that it would be impossible to distinguish a run of Algorithm 3 from a run of Algorithm 1 with the Gumbel values sorted. This allows us to use any choice of $partition$ with a run of Algorithm 1 to analyze the distribution of $\max\{G_k \,|\, X_k \in B\}$. More precisely

1. We argue that the top $n$ Gumbels of Algorithm 1 are distributed as in Algorithm 3 regardless of $partition$. That is, if $[i]$ is the index of the $i$th largest Gumbel, then for $1 \leq n \leq |\Omega|$

$$G_{[1]} \sim \text{Gumbel}(\log \mu(\Omega))$$
$$G_{[k]} \sim \text{TruncGumbel}(\log \mu(\Omega_{k-1}), G_{[k-1]}) \text{ for } 1 < k \leq n$$
$$X_{[k]} \sim \mathbf{1}(x \in \Omega_k)\exp(\phi(x))/\mu(\Omega_k) \text{ for } 1 \leq k \leq n$$

This implies that the distribution over $\{\max\{G_k \mid X_k \in B\} \mid B \subseteq \Omega\}$ is invariant under the choice of *partition* function in Algorithm 1.

2. We derive the following for a specific choice of *partition*

$$\max\{G_k \mid X_k \in B\} \sim \text{Gumbel}(\log \mu(B))$$
$$\max\{G_k \mid X_k \in B^c\} \sim \text{Gumbel}(\log \mu(B^c))$$
$$\max\{G_k \mid X_k \in B\} \perp \max\{G_k \mid X_k \in B^c\}$$

By the previous result, this is the distribution for any choice of *partition* (provided it doesn't produce immeasurable sets) giving us conditions 1. and 2. of Definition 1. Condition 3. is easily satisfied.

This proves the existence of the Gumbel process.

**Equivalence Under** *partition*

We will proceed to show that the distribution over $\{\max\{G_k \mid X_k \in B\} \mid B \subseteq \Omega\}$ is invariant under the choice of *partition* function. To do so we argue that the top $n$ Gumbels from Algorithm 1 all have the distribution from Algorithm 3. That is, if $[k]$ is the index of the $k$th smallest Gumbel in the tree from Algorithm 1 and $\Omega_k = \Omega - \cup_{i=1}^{k-1}\{X_{[k]}\}$. Then for all $n \leq |\Omega|$

$$G_{[1]} \sim \text{Gumbel}(\log \mu(\Omega))$$
$$G_{[k]} \sim \text{TruncGumbel}(\log \mu(\Omega_{k-1}), G_{[k-1]}) \text{ for } 1 < k \leq n$$
$$X_{[k]} \sim \mathbf{1}(x \in \Omega_k)\exp(\phi(x))/\mu(\Omega_k) \text{ for } 1 \leq k \leq n$$

Notice that whenever $\mu(\Omega_k) = \mu(\Omega_{k+1})$ we can omit the removal of $X_k$ and still have the same distribution. In the case of continuous $\mu$ we can completely omit all removals and set $\Omega_k = \Omega$.

*Proof.* We proceed by induction. For $n = 1$, clearly

$$G_{[1]} = G_1 \sim \text{Gumbel}(\log \mu(\Omega))$$
$$X_{[1]} = X_1 \sim \exp(\phi(x))/\mu(\Omega)$$

Now for $1 < n \leq |\Omega|$, consider the the top $n$ nodes from a single realization of the process. Let $[<k] = \{[1],, \ldots, [k-1]\}$, the indices of the first $k-1$ Gumbels. By the induction hypothesis we know their distribution and they form a partial tree of the completely realized tree. Our goal is to show that

$$G_{[n+1]} \sim \text{TruncGumbel}(\log \mu(\Omega_{n+1}), G_{[n]})$$
$$X_{[n+1]} \sim \mathbf{1}(x \in \Omega_{n+1})\exp(\phi(x))/\mu(\Omega_{n+1})$$

The boundary of the max partial tree are the nodes $i$ that are on the Queue and have not been expanded. We know that conditioned on $[<n+1]$ that $G_{[n+1]} = \max_{i \notin [<n+1]} G_i$ will come from this boundary, i.e. $G_{[n+1]} = \max_{i \in boundary} G_i$. The first step is to realize that the sets $B_i$ on the boundary of the max partial tree form a partition of $\Omega_{n+1}$. If $g_{[n+1]} = \max_{i \in boundary} g_i$ and $p_i$ is the parent of node $i$, then

$$p(\forall i \in boundary, G_i = g_i, G_{[1]} = g_{[1]}, \ldots, G_{[n]} = g_{[n]} \mid [<n+1])$$

$$\propto \prod_{i \in boundary} f_{\log B_i}(g_i)\mathbf{1}(g_{p_i} > g_i) \prod_{k=1}^{n} f_{\log \mu(\Omega_k)}(g_{[k]})\mathbf{1}\left(g_{[k]} > g_{[k+1]}\right)$$

Because products of indicator functions are like intersections

$$p(\forall i \in boundary, G_i = g_i \mid G_{[1]} = g_{[1]}, \ldots, G_{[n]} = g_{[n]}, [<n+1]) \propto \prod_{i \in boundary} f_{\log \mu(B_i)}(g_i)\mathbf{1}\left(G_{[n]} > g_i\right)$$

In other words, the boundary Gumbels are independent and $G_i \sim \text{TruncGumbel}(\log \mu(B_i), G_{[n]})$. Notice that the subsets of the boundary form a complete partition of $\Omega_{n+1}$, thus we get

$$G_{[n+1]} \sim \text{TruncGumbel}(\log \mu(\Omega_{n+1}), G_{[n]})$$

The location $X_{[n+1]}$ has the following distribution:

$$[n+1] \sim \mathrm{argmax}\{G_i \,|\, i \in boundary\}$$

$$X_{[n+1]} \sim \mathbf{1}\big(x \in B_{[n+1]}\big) \exp(\phi(x))/\mu(B_{[n+1]})$$

Again, because the $B_i$ is a partition of $\Omega_{n+1}$, this is a mixture distribution in which subsets $B_{[n+1]}$ are sampled with probability $\mu(B_{[n+1]})/\mu(\Omega_{n+1})$ and then $X_{[n+1]}$ is sampled from $\mathbf{1}\big(x \in B_{[n+1]}\big) \exp(\phi(x))/\mu(B_{[n+1]})$. Thus,

$$X_{[n+1]} \sim \mathbf{1}(x \in \Omega_{n+1}) \exp(\phi(x))/\mu(\Omega_{n+1})$$

and by the independence of the max and argmax we get that $X_{[n+1]}$ is independent of $G_{[n+1]}$. $\quad\square$

### Joint Marginal of Max-Gumbels in $B$ and $B^c$

Because the joint distribution over the entire collection $\{\max\{G_k \,|\, X_k \in B\} \,|\, B \subseteq \Omega\}$ is the same regardless of *partition*, this implies that the joint of $\max\{G_k \,|\, X_k \in B\}$ and $\max\{G_k \,|\, X_k \in B^c\}$ for any specific choice of partition is indeed the joint marginal for *any partition*. In particular we show

$$\max\{G_k \,|\, X_k \in B\} \sim \mathrm{Gumbel}(\log \mu(B))$$
$$\max\{G_k \,|\, X_k \in B^c\} \sim \mathrm{Gumbel}(\log \mu(B^c))$$
$$\max\{G_k \,|\, X_k \in B\} \perp \max\{G_k \,|\, X_k \in B^c\}$$

*Proof.* Consider the *partition* that first partitions $\Omega$ into $B$ and $B^c$. In this case we consider the distribution over $G_B = \max\{G_k \,|\, X_k \in B\}$ and $G_{B^c} = \max\{G_k \,|\, X_k \in B^c\}$ in Algorithm 1. If $X_1 \in B$, then $G_B = G_1$ and $G_{B^c} = G_3$. Otherwise $G_{B^c} = G_1$ and $G_B = G_2$. Thus, $G_B > G_{B^c}$ iff $X_1 \in B$. Using this knowledge we can split the distribution over $G_B$ and $G_{B^c}$ into two events.

$p(G_B = g_b, G_{B^c} = g_{B^c})$
$= p(G_B = g_b, G_{B^c} = g_{B^c} \,|\, G_B > G_{B^c})p(G_B > G_{B^c}) + p(G_B = g_b, G_{B^c} = g_{B^c} \,|\, G_B \leq G_{B^c})p(G_B \leq G_{B^c})$
$= f_{\log \mu(\Omega)}(g_B)\dfrac{f_{\log \mu(B^c)}(g_{B^c})\mathbf{1}(g_B > g_{B^c})}{F_{\log \mu(B^c)}(g_B)}\dfrac{\mu(B)}{\mu(\Omega)} + f_{\log \mu(\Omega)}(g_{B^c})\dfrac{f_{\log \mu(B)}(g_B)\mathbf{1}(g_B \leq g_{B^c})}{F_{\log \mu(B)}(g_{B^c})}\dfrac{\mu(B^c)}{\mu(\Omega)}$
$= f_{\log \mu(B)}(g_B)f_{\log \mu(B^c)}(g_{B^c})\mathbf{1}(g_B > g_{B^c}) + f_{\log \mu(B^c)}(g_{B^c})f_{\log \mu(B)}(g_B)\mathbf{1}(g_B \leq g_{B^c})$
$= f_{\log \mu(B)}(g_B)f_{\log \mu(B^c)}(g_{B^c})$

This is the density of two independent Gumbels with locations $\log \mu(B)$ and $\log \mu(B^c)$. This proves our result. $\quad\square$

## Analysis of A* Sampling

This section deals with the correctness and termination of A* sampling. We exclusively analyze the continuous version of A* sampling. Recall we have two continuous measures $\mu(B) = \int_{x \in B} \exp(\phi(x))$ and $\nu(B) = \int_{x \in B} \exp(i(x))$ such that we can decompose $\phi(x)$ in a tractable $i(x)$ and intractable but boundable component $o(x)$.

$$\phi(x) = i(x) + o(x)$$

### Termination of A* Sampling

In this section we argue that A* sampling terminates with probability one by bounding it with the runtime of global-bound A* sampling. We analyze global-bound A* more closely.

Consider running A* with two different sets of bounds on the same realization of the Gumbel process. The returned sample, the final lower bound, and the split chosen for any region will be the same. The only thing that changes is the set of nodes in the tree that are explored. Let $U_{A^*}(B) = G_k + M(B)$ be the upper bound at node $B$ for A* and $U(B) = G_k + M$ be the upper bound at node $B$ for global-bound A*. Because these algorithms are searching on the same realization we assume that

**Algorithm 4** Global-Bound A* Sampling

---

**input** log density $i(x)$, difference $o(x)$, bounding function $M(B)$
  $(LB, \ X^*, \ k) \leftarrow (-\infty, \ \text{null}, \ 1)$
  $G_1 \sim (\text{Gumbel}(\log \nu(\mathbb{R}^d))$
  $X_1 \sim \exp(i(x))/\nu(\mathbb{R}^d))$
  $M \leftarrow M(\mathbb{R}^d)$
  **while** $LB < G_k + M$ **do**
    $LB_k \leftarrow G_k + o(X_k)$
    **if** $LB < LB_k$ **then**
      $LB \leftarrow LB_k$
      $X^* \leftarrow X_k$
    $k \leftarrow k + 1$
    $G_k \sim \text{TruncGumbel}(\log \nu(\mathbb{R}^d), G_{k-1})$
    $X_k \sim \exp(i(x))/\nu(\mathbb{R}^d)$
**output** $(LB, X^*)$

---

$U(B) \geq U_{A^*}(B)$. Let $LB$ be the final lower bound—the optimal node. Because $U(B) \geq U_{A^*}(B)$, we know that global-bound A* visits at least the nodes for which

$$U_{A^*}(B) \geq LB$$

Finally, A* never visits nodes for which

$$U_{A^*}(B) < LB$$

So, A* cannot visit a node that global-bound A* never visits. Thus, if global-bound A* terminates with probability one, then so does A*. We now analyze the run time of global-bound A* more closely and discover a parallel with rejection sampling.

**Termination of Global-Bound A* Sampling**

If a global bound $M \geq o(x)$ is reused at every node in A* sampling, then it takes on a particularly simple form, Algorithm 4; no queue is needed, and it simplifies to a search over the stream of $(G_k, X_k)$ values from the In-Order construction (Alg. 3). Global-bound A* sampling is equivalent to rejection sampling. In particular, both rejection and A* sampling with constant bounds terminate after $k$ iterations with probability

$$(1 - \rho)^{k-1}\rho$$

where $\rho = \mu(\mathbb{R}^d)(\exp(M)\nu(\mathbb{R}^d))^{-1}$.

Rejection terminates with this probability, because the termination condition is independent for each iteration. Thus, the distribution over the number of iterations is a geometric with probability:

$$
\begin{aligned}
\rho &= \mathrm{P}\left(U \leq \exp(o(X) - M)\right) \\
&= \mathbb{E}[\exp(o(X) - M)] \\
&= \frac{1}{\exp(M)} \int_{x \in \mathbb{R}^d} \exp(\phi(x) - i(x)) \frac{\exp(i(x))}{\nu(\mathbb{R}^d)} \\
&= \frac{\mu(\mathbb{R}^d)}{\exp(M)\nu(\mathbb{R}^d)}
\end{aligned}
$$

That global-bound A* terminates with this probability is interesting, because the termination condition is not independent from the history of the Gumbel values. Nonetheless, the distribution over iterations is memoryless. Consider the stream of values $(G_k, X_k)$,

$$
\begin{aligned}
G_1 &\sim \text{Gumbel}(\log \mu(\mathbb{R}^d)) \\
G_k &\sim \text{TruncGumbel}(\log \mu(\mathbb{R}^d), G_{k-1}) \text{ for } k > 1 \\
X_k &\sim \exp(\phi(x))/\mu(\mathbb{R}^d)
\end{aligned}
$$

Global-bound A* terminates when $\max_{1 \le i \le k}\{G_i + o(X_i)\} \ge G_{k+1} + M$. In order to show that the distribution of $k$ is geometric we need simply to show that

$$P\left(\max_{1 \le i \le k}\{G_i + o(X_i) - M\} < G_{k+1}\right) = (1 - \rho)^k$$

*Proof.* First we show that the finite differences $D_k = G_{k+1} - G_k$ are mutual independent and

$$-D_k \sim \text{exponential}(k)$$

Inspecting the joint pdf for $G_i$ with $1 \le i \le k+1$

$$f(g_1, \ldots, g_{k+1}) = f_\mu(g_1) \prod_{i=2}^{k+1} \frac{f_\mu(g_i)}{F_\mu(g_{i-1})} \mathbf{1}(g_i \le g_{i-1})$$

$$= \left(\prod_{i=1}^{k+1} e_\mu(g_i)\right) F_\mu(g_{k+1}) \prod_{i=2}^{k+1} \mathbf{1}(g_i \le g_{i-1})$$

We proceed with an inductive argument. First the base, with $k = 1$, so for $d_1 < 0$

$$p(D_1 = d_1) = \int_{g_1 = -\infty}^{\infty} e_\mu(g_1) F_\mu(g_1 + d_1)$$

$$= \exp(d_1) \int_{g_1 = -\infty}^{\infty} e_\mu(g_1) F_\mu(g_1)$$

$$= \exp(d_1)$$

Now by induction

$$P(D_1 \le d_1, \ldots, D_k \le d_k) = \int_{g_1 = -\infty}^{\infty} e_\mu(g_1) \int_{g_2 = -\infty}^{g_1 + d_1} e_\mu(g_2) \ldots \int_{g_{k+1} = -\infty}^{g_k + d_k} f_\mu(g_{k+1})$$

$$= \int_{g_1 = -\infty}^{\infty} e_\mu(g_1) \int_{g_2 = -\infty}^{g_1 + d_1} e_\mu(g_2) \ldots \int_{g_k = -\infty}^{g_{k-1} + d_{k-1}} e_\mu(g_k) F_\mu(g_k + d_k)$$

$$= \exp(d_k) P(D_1 \le d_1, \ldots, D_{k-1} \le d_{k-1} + d_k)$$

$$= \exp(d_k) \exp((k-1)(d_{k-1} + d_k)) \prod_{i=1}^{k-2} \exp(i d_i)$$

$$= \prod_{i=1}^{k} \exp(i d_i)$$

Now we proceed to show that

$$P\left(\max_{1 \le i \le k}\{G_i + o(X_i) - M\} < G_{k+1}\right) = (1 - \rho)^k$$

First, let $Y = o(X) - M$ with $X \sim \exp(\phi(x))/\mu(\mathbb{R}^d)$ have PDF $h(y)$ and CDF $H(y)$. Notice that $P(Y \le 0) = 1$ and $\mathbb{E}[\exp(Y)] = \rho$. We rewrite the event of interest as a joint event in the finite differences of the Gumbel chain and then use the independence of the $X_i$

$$P\left(\max_{1 \le i \le k}\{G_i + o(X_i) - M\} < G_{k+1}\right) = P\left(\{G_{k+1} - G_i > o(X_i) - M\}_{i=1}^k\right)$$

$$= \mathbb{E}\left[P\left(\left\{\sum_{j=i}^{k} D_j > o(X_i) - M\right\}_{i=1}^k \middle| \{D_j\}_{j=1}^k\right)\right]$$

$$= \mathbb{E}\left[\prod_{i=1}^{k} H\left(\sum_{j=i}^{k} D_j\right)\right]$$

looking more closely at this event

$$\mathbb{E}\left[\prod_{i=1}^{k} H\left(\sum_{j=i}^{k} D_j\right)\right] = \int_{d_k=-\infty}^{0}\int_{d_{k-1}=-\infty}^{0}\cdots\int_{d_1=-\infty}^{0}\prod_{i=1}^{k} H\left(\sum_{j=i}^{k} d_j\right)\prod_{i=1}^{k} i\exp(id_i)$$

$$= k!\int_{d_k=-\infty}^{0}\int_{d_{k-1}=-\infty}^{0}\cdots\int_{d_1=-\infty}^{0}\prod_{i=1}^{k} H\left(\sum_{j=i}^{k} d_j\right)\exp\left(\sum_{j=i}^{k} d_j\right)$$

Now we do a sequence of tricky substitutions, $r_i = d_i + \sum_{j=i+1}^{k} d_j$ from $i = 1$ to $k$, and find that this integral equals

$$k!\int_{r_k=-\infty}^{0}\int_{r_{k-1}=-\infty}^{r_k}\cdots\int_{r_1=-\infty}^{r_2}\prod_{i=1}^{k} H\left(r_i\right)\exp\left(r_i\right)$$

Notice that this is basically an infinite triangle over a function that is symmetric in the $r_i$. Since it's multiplied by $k!$ it is equal to the sum over all the permutations of the $r_i$, which ends up giving us an infinite cube:

$$k!\int_{r_k=-\infty}^{0}\int_{r_{k-1}=-\infty}^{0}\cdots\int_{r_1=-\infty}^{0}\prod_{i=1}^{k} H\left(r_i\right)\exp\left(r_i\right) = \left(\int_{r=-\infty}^{0} H\left(r\right)\exp\left(r\right)\right)^{k}$$

All that remains is to evaluate $\int_{r=-\infty}^{0} H\left(r\right)\exp\left(r\right)$:

$$\int_{r=-\infty}^{0} H\left(r\right)\exp\left(r\right) = \int_{r=-\infty}^{0}\int_{y=-\infty}^{r} h\left(y\right)\exp\left(r\right)$$

$$= \int_{y=-\infty}^{0}\int_{r=y}^{0} h\left(y\right)\exp\left(r\right)$$

$$= \int_{y=-\infty}^{0} h(y)(1 - \exp\left(y\right))$$

$$= 1 - \rho$$

and that completes the proof. $\qquad\square$

**Partial Correctness of A* Sampling**

In this section we show that given termination the distribution returned by A* is correct. This depends only on the linearity result about the Gumbel process. We prove the linearity by using the auxiliary $\mathcal{G}_\nu$ to measure the bounded difference $o(x)$.

At termination A* returns $LB$ and $X^*$:

$$LB = \max\{G_k + o(X_k)\}$$
$$X^* = \operatorname{argmax}\{G_k + o(X_k)\}$$

where $(G_k, X_k)$ are a stream of Gumbels and locations obtained from running Algorithm 3 with measure $\nu$. Our goal is to show that $LB \sim \operatorname{Gumbel}(\log\mu(\Omega))$ and $X^* \sim \exp(\phi(x))/\mu(\Omega)$. Thus, by Definition 1 it is sufficient to show that $\{\max\{G_k + o(X_k)\,|\,X_k \in B\}\,|\,B \subseteq \mathbb{R}^d\}$ is a Gumbel process $\mathcal{G}_\mu$.

*Proof.* The correctness of the construction for $\mathcal{G}_\nu$ implies the consistency and independence requirements. Thus we need only to verify the marginal of $\max\{G_k + o(X_k)\,|\,X_k \in B\}$ is $\operatorname{Gumbel}(\log\mu(B))$. To see why this is the case, consider a partition $p_1$, ..., $p_n$ of the range of $o(x)$ and let

$$P_j = \{x\,|\,p_{j-1} < o(x) \le p_j\}$$

with $p_0 = -\infty$ and $p_n = M$. Then

$$\max\{G_k + o(X_k)\,|\,X_k \in B\} = \max_j\{\max\{G_k + o(X_k)\,|\,X_k \in B \cap P_j\}\}$$

thus

$$\max_j\{\max\{G_k \mid X_k \in B \cap P_j\} + p_{j-1}\} \leq \max\{G_k + o(X_k) \mid X_k \in B\} \leq \max_j\{\max\{G_k \mid X_k \in B \cap P_j\} + p_j\}$$

$\{G_k \mid X_k \in B \cap P_j\} \sim \text{Gumbel}(\log \nu(B \cap P_j))$, because $G_k$ and $X_k$ are samples from the process $G_i(B)$. Thus,

$$\max_j\{\max\{G_k \mid X_k \in B \cap P_j\} + p_j\} \sim \text{Gumbel}(\log \sum_j \exp(\log \nu(B \cap P_j) + p_j))$$

similarly

$$\max_j\{\max\{G_k \mid X_k \in B \cap P_j\} + p_{j-1}\} \sim \text{Gumbel}(\log \sum_j \exp(\log \nu(B \cap P_j) + p_{j-1}))$$

we see that

$$\sum_j \nu(B \cap P_j)\exp(p_{j-1}) \to \int_{x \in B} \exp(i(x))\exp(o(x)) \leftarrow \sum_j \nu(B \cap P_j)\exp(p_j)$$

as the partition gets finer. Since $\int_{x \in B} \exp(i(x))\exp(o(x)) = \mu(B)$ we get that

$$\max_j\{\max\{G_k \mid X_k \in B \cap P_j\} + p_j\} \xrightarrow{d} \text{Gumbel}(\log \mu(B))$$

$$\max_j\{\max\{G_k \mid X_k \in B \cap P_j\} + p_{j-1}\} \xrightarrow{d} \text{Gumbel}(\log \mu(B))$$

Thus the distribution of $\max\{G_k + o(X_k) \mid X_k \in B\}$ must be $\text{Gumbel}(\log \mu(B))$ and we're done.
$\square$

**Explanation of Results from Section 6.2**

Consider running A$^*$ with two different sets of bounds on the same realization of the Gumbel process. The returned sample, the final lower bound, and the split chosen for any region will be the same. The only thing that changes is the set of regions that are explored. Let $U_1(B)$ be an optimal bound and $U_2(B)$ a suboptimal bound of region $R$. The question is how many more regions are explored by using $U_2$ instead of $U_1$. Let $R^*$ be the region producing the returned sample. Once this region is explored, the lower bound reaches its final value and A$^*$ will only explore regions with $U_2(B) > LB$. These regions will be new, have been unexplored by $U_1$, if $U_1(B) < LB$. Before $R^*$ is explored, A$^*$ will only explore regions with $U_2(B) > U_2(R^*)$. Since $U_2(R^*) > LB$, the condition that determines whether a region $R$ is new is $U_2(B) > LB > U_1(B)$.

Suppose $R$ is explored using $U_1$, but its descendants are not. How many of its descendants are explored using $U_2$? For the bounds we consider in Section 6.2, the suboptimality of $U_2$ is proportional to the region width. Let $B_d$ be the deepest descendant explored using $U_2$, and let $n$ be the width of $R$ divided by $B_d$. By the assumption on the suboptimality of bounds being proportional to region width, we have $(U_2(B) - U_1(B))/n > U_2(B_d) - U_1(B_d)$, and thus $(U_2(B) - U_1(B))/n + U_1(B_d) > LB$. This implies that $n$ is bounded by a linear function of the suboptimality $U_2(B) - U_1(B)$. The balanced nature of the splitting process suggests that $B_d$ has depth $\log_2(n)$ with high probability, so $n$ also bounds the total number of explored descendants. Therefore the total number of additional regions explored by $U_2$ is linear in the total suboptimality of the bounds of regions explored by $U_1$.

When the log-likelihood is a sum of $n$ terms and we apply a constant bound to each term, the suboptimality of the total bound grows linearly with $n$. However, under the conditions of the Bernstein-von Mises theorem, the posterior will concentrate around a peak of width $O(n^{-1/2})$. This shrinks the width of the significant regions, reducing the suboptimality of an explored region to $O(\sqrt{n})$. This is the trend we see in the plot. If we apply a linear bound to each term, the suboptimality is $O((\text{width})^2)$ per term by the Taylor remainder theorem, and overall $O(n(\text{width})^2)$ which is constant in $n$ for regions around the peak. Therefore we expect linear termwise bounds to explore a constant multiple of the number of regions explored by optimal bounds.

**Bounding the Cauchy Log Likelihood**

To perform inference in the Bayesian Robust Regression experiment, we need to upper bound $\max_{\boldsymbol{w} \in [\underline{\boldsymbol{w}}, \overline{\boldsymbol{w}}]} \mathcal{L}(\boldsymbol{w})$ for each $[\underline{\boldsymbol{w}}, \overline{\boldsymbol{w}}]$ encountered in the search tree. For each region, we compute the bound in two steps. First, for each $n$, compute the minimum and maximum possible values of $d_n = \boldsymbol{w}^\mathsf{T} \boldsymbol{x}_n - y_n$ using interval arithmetic [16], yielding $\underline{d_n}$ and $\overline{d_n}$. For each $n$, we then construct a quadratic bound on the Cauchy likelihood term $C(d) = -\log(1 + d^2)$ that is guaranteed to be an upper bound so long as $d_n \in [\underline{d_n}, \overline{d_n}]$. The bound is referred to as $B_n(d_n)$ and takes the form $B_n(d_n) = a_n d_n^2 + b_n d_n + c_n$.

The second derivative of the Cauchy likelihood $C''(d) = \frac{4d^2}{(d^2+1)^2} - \frac{2}{d+1}$ changes sign only twice, at $-1$ and $1$. Outside of $[-1, 1]$, $C''(d)$ is positive (i.e., $C$ is convex), and inside it is negative (i.e., $C$ is concave). If an interval $[\underline{d_n}, \overline{d_n}]$ is fully in a convex region, then we use a simple linear bound of the line that passes through $(\underline{d_n}, C(\underline{d_n}))$ and $(\overline{d_n}, C(\overline{d_n}))$. If an interval $[\underline{d_n}, \overline{d_n}]$ is fully in a concave region and does not contain $0$, then we use a linear bound that is tangent to $C$ at the midpoint of $[\underline{d_n}, \overline{d_n}]$. If the interval contains any of $\{-1, 0, 1\}$ then we use a quadratic bound. If the interval contains one of $-1$ or $1$, then we expand the interval to include $0$ and then proceed. To compute the bound, we fix the bound function to have $B_n(0) = C(0)$ and $B'_n(0) = C'(0)$. Since the bound is quadratic, its second derivative $2a_n$ is constant. We set this constant to be the most negative value that ensures that $B_n(d)$ is a valid bound over all of $[\underline{d}, \overline{d}]$. Concretely, the quadratic bound $B(d) = ad^2 + bd + c$ is constructed as follows. For each endpoint $d_{end} \in \{\underline{d_n}, \overline{d_n}\}$ that is not equal to $0$, compute

$$a = \frac{C(d_{end}) - C(0)}{d_{end}^2}. \tag{7}$$

and choose the largest computed $a$. Finally, solve for $b$ and $c$ to ensure that the derivative and value of the bound match $C$ at $0$:

$$b = f'(d_0) - 2ad_0 \tag{8}$$
$$c = f(d_0) - ad_0^2 - bd_0. \tag{9}$$

**Regression Experiment Priors**

All parameters were given uniform priors. The ranges are as follows:

$y = a \exp(-b \,|x - c|^d) + e$

    a   [.1, 5]
    b   [.5, 5]
    c   [-5, 5]
    d   [.1, 5]
    e   [.1, 5]

$y = a \sin(bx + c) + d \sin(ex + f)$

    a   [-5, 5]
    b   [-5, 5]
    c   [-5, 5]
    d   [-5, 5]
    e   [-5, 5]
    f   [-5, 5]

$y = a(x - b)^2 / ((x - b)^2 + c^2)$

    a   [-5, 5]
    b   [-5, 5]
    c   [-5, 5]

$y = x \frac{\cos(a)(x \sin(a) + \sqrt{x^2 \sin(a)^2 + 2bc})}{b}$

a    $[0.01, \pi - .01]$
b           $[.1, 5]$
c           $[0, 5]$

$$y = ax(x - b)(c - x)^d$$

a    $[.01, 1]$
b    $[.5, 1]$
c    $[2, 3]$
d    $[.1, 1]$