[Reviews · NeurIPS 2014]

Submitted by Assigned_Reviewer_11

This paper introduces a new approach to sampling from continuous probability distributions. The method extends prior work on using a combination of Gumbel perturbations and optimization to the continuous case. This is technically challenging, and they devise several interesting ideas to deal with continuous spaces, e.g. to produce an exponentially large or even infinite number of random variables (one per point of the continuous/discrete space) with the right distribution in an implicit way. Finally, they highlight an interesting connection with adaptive rejection sampling. Some experimental results are provided and show the promise of the approach.

The paper is well written (I have a few suggestions below) and generally easy to read (Figure 1 is nice and very helpful to understand the algorithm). This is very novel work where the authors explore an interesting combination of ideas from computer science and traditional AI (like A* search) with advanced statistics. The paper deals with a very important problem which is very relevant to NIPS.

The experimental evaluation is a bit limited (and only on synethetic data) but I think it might be good enough as an initial step. One issue is that the algorithm doesn't seem to be able to handle high-dimensional spaces well, but that is of course very hard. One concern I have is that the performance metric used in the experimental section is "likelihood evaluations per sample". What about runtime? Are there overheads in using A* sampling?

Other comments:

It would be good to specify clearly what is the input and what is the output of the algorithms reported as pseudocode (Algorithm 1 and 2).

Line 177. I would expand the first paragraph of section 4. The decomposition in tractable and intractable parts is key. The results in the previous sections obviously don't make any sense if we can already sample and integrate. I would try to say more about what the decomposition means, maybe mentioning some of the examples used later.

Explanation in section 6.2 is a bit confusing (especially the last paragraph is rather obscure)

In 6.3, why is it compared to rejection sampler and not with OS*? Is it so slow that can only do 3 points and 5 runs?

How does one deal with ties? If the argmax contains more than one element, do we have to sample uniformly from the set or is it sufficient to output one?
I am guessing in principle it only happens with probability zero so it might not be an issue, but since you are generating an exponentially large number of random variable using finite precision arithmetic can it still happen?

It's unclear what the probability is conditioned on in Eq (3)

Line 85-86: at that point it's unclear what it means that max and argmax are independent

line 4 of pseudocode: are there technical assumptions for the max to be defined (as opposed to a supremum)
Summary: A nice paper introducing a new technique to sample from continuous distribution using a clever combinations of ideas from AI and statistics.

Submitted by Assigned_Reviewer_13

Brief Description
This paper discusses how to perform exact sampling from a target distribution via the formulation of an optimisation problem. In particular, the idea discussed originates from sampling of discrete distributions and is formulated based on the Gumbel process. It is claimed that the A-star sampling method proposed based on A-star search is more efficient than adaptive rejection sampling.

Quality
The paper is interesting for this audience and well presented.

Clarity
Generally the paper is well written- however it would benefit from a more in-depth discussion on suitable problems that can be addressed by this proposed sampling method. In addition, the figures are too small to view and the maximum dimension illustrated is only 4 dimensional. This is somewhat surprising as it would be good to see the performance on very high dimensional settings.

Orginality
This paper extends existing techniques and merges several streams of work to make a good contribution.

Significance
Relevant to many groups in this audience.
Summary: The paper is interesting and well presented - I would recommend more discussion on the pros and cons of the proposed sampling method as well as extended discussion on the adaptive rejection sampler and OS sampler discussed. Comparison with these would also be interesting.

Submitted by Assigned_Reviewer_40

Summary:
This paper introduces a sampling algorithm based on the Gumbel-max trick and A* search for continuous spaces. The Gumbel-Max trick adds perturbations to an energy function and after applying argmax, results in exact samples from the Gibbs distribution. While this applies to discrete spaces, this paper extends this idea to continuous spaces using the upper bounds on the infinitely many perturbation values.

Quality:
Paper has a very solid motivation.
The quality of the paper is very good. All claims have proofs in the appendix and experimental results compare the main properties. The weakness of their method is also explained in the discussion.

Clarity:
The paper is well written and clear. The two main algorithms are easy to read and explained in the text. The reader does not require reading about the discrete case as it is explained before the continuous claims.

Originality:
This work is a clear extension from the work on applying Gumbel-max on discrete spaces. The main point is how Gumbel-max can be used for continuous spaces, which is by the Gumbel process already defined in the literature.
The heap construction of the Gumbel process is a new idea to allow extending it to the continuous case as shown in Algorithm 1 and sampling in the descending order will not change the max value.
The A* sampling algorithm is then used to draw samples from the continuous distribution. It is also compared with OS* is considered for continuous spaces. The A* algorithm is clear and shows how upper and lower bounds are set for the sampling algorithm. Properties of this algorithm are defined and proven in the appendix.

Significance:
In relation to the empirical results, the search tree regions are hyper rectangular which is too restrictive. Ideas to extend the ideas for higher dimensions need to be mentioned. Although this paper is considered an extension to its discrete version, it has novel insights in how to use in continuous spaces.
Summary: An extension to using the Gumbel-max trick for discrete spaces, this paper uses the Gumbel process for continuous spaces and introduces the A* sampling algorithm for low dimensional continuous space problems.
Author Feedback
Author rebuttal: We thank the reviewers for their time and feedback. The writing suggestions are appreciated and will be incorporated into a revised version.

REVIEWER 11

> R11: the performance metric used in the experimental section is "likelihood evaluations per sample". What about runtime? Are there overheads in using A* sampling?

The overhead in A* sampling is (up to) two O(log n) operations per node expansion, which are required to push new nodes onto the priority queue. A naive implementation of OS* would incur an O(n) operation for each proposal due to having to sample a region, but this could be made logarithmic by using a tree structure or perhaps better using more sophisticated approaches (see e.g. [A, cited below]). Note however that the n in A* may be smaller than the n in OS* because of pruning within A*.

Having said this, we implemented the naive version of region sampling within OS*, and our implementations are not particularly optimized at a low level (they are in python), so we felt that bound and likelihood evaluations were a more meaningful way of reporting the results.

[A] http://theory.stanford.edu/~matias/papers/rv.pdf

> R11: In 6.3, why is it compared to rejection sampler and not with OS*?

The point we wanted to make in this experiment was just that A* sampling can use bounds that are derived automatically by a branch and bound library. The rejection baseline is there to establish the difficulty of the problems, e.g., to show that we hadn’t chosen a setting of the observation noise that makes the problem trivial. We’re happy to add comparison to OS*.

> R11: Is it so slow that can only do 3 points and 5 runs?

Having a small number of points can actually make the problems more difficult, because it is more likely to lead to multimodality in the posterior. Peakiness of the posterior is controlled orthogonally by varying the observation noise. We designed these experiments so that it kept generating more difficult problems until it started taking a long time to run.

> R11: How does one deal with ties? If the argmax contains more than one element, do we have to sample uniformly from the set or is it sufficient to output one? I am guessing in principle it only happens with probability zero so it might not be an issue, but since you are generating an exponentially large number of random variable using finite precision arithmetic can it still happen?

This does happen with probability 0 so isn’t an issue when using real numbers, but the algorithm would just output one from the set. With finite precision arithmetic, we may get with very low probability some k,k’ in Algorithm 2 such that Xk and Xk’ are both in the argmax set, i.e., o(Xk) + Gk = o(Xk’) + Gk’. In this case, Algorithm 2 will choose the first of the k’s that is generated by the algorithm.

> R11: line 4 of pseudocode: are there technical assumptions for the max to be defined (as opposed to a supremum)

Thank you for pointing this out. We were sloppy with the use of max. The actual requirement is just that M’s in Algorithm 2 are upper bounds, and we need one other technical condition that branch and bound algorithms generally need, which is the upper bound becomes a supremum as the region shrinks to be infinitely small. We will clarify this.

REVIEWER 13

> R13: I would recommend extended discussion on the adaptive rejection sampler and OS sampler discussed. Comparison with these would also be interesting.

We spend a full section (Section 5) on the relationship to adaptive rejection sampling (and in particular OS*). We compare to OS* experimentally in Sec 6.1 and 6.4.

REVIEWER 40

> R40: the maximum dimension illustrated is only 4 dimensional

The problems in Fig 2d are up to six dimensional, but as we say in the discussion, we do not claim to solve the (very hard) problem of exact sampling in high dimensions.

> R40: Ideas to extend the ideas for higher dimensions need to be mentioned.

We provide ideas in the discussion. See L415 when we suggest approximations similar to what Hazan & Jaakkola [3] did in the discrete case, and L431 (referencing citation [21]) for an inspiring example of branch and bound search that works in high dimensions by exploiting a particular type of conditional independence structure.